# Hydrogel-Based Systems as Smart Food Packaging: A Review

**DOI:** 10.3390/polym17081005

**Published:** 2025-04-08

**Authors:** Beata Niemczyk-Soczynska, Paweł Łukasz Sajkiewicz

**Affiliations:** Laboratory of Polymers & Biomaterials, Institute of Fundamental Technological Research, Polish Academy of Sciences, Pawińskiego 5B St., 02-106 Warsaw, Poland

**Keywords:** hydrogels, bio-based polymers, active packaging, intelligent packaging, food packaging

## Abstract

In recent years, non-degradable petroleum-based polymer packaging has generated serious disposal, pollution, and ecological issues. The application of biodegradable food packaging for common purposes could overcome these problems. Bio-based hydrogel films are interesting materials as potential alternatives to non-biodegradable commercial food packaging due to biodegradability, biocompatibility, ease of processability, low cost of production, and the absorption ability of food exudates. The rising need to provide additional functionality for food packaging has led scientists to design approaches extending the shelf life of food products by incorporating antimicrobial and antioxidant agents and sensing the accurate moment of food spoilage. In this review, we thoroughly discuss recent hydrogel-based film applications such as active, intelligent packaging, as well as a combination of these approaches. We highlight their potential as food packaging but also indicate the drawbacks, especially poor barrier and mechanical properties, that need to be improved in the future. We emphasize discussions on the mechanical properties of currently studied hydrogels and compare them with current commercial food packaging. Finally, the future directions of these types of approaches are described.

## 1. Introduction

Nowadays, plastics derived from fossil fuels, mostly from crude oil, dominate the food packaging market and have been widely used worldwide for several decades [1]. This is due to their outstanding packaging attributes, such as high gas and water barriers, as well as favorable mechanical properties. Their other advantages are low costs and ease of production and processing, as well as fast-growing technological innovations in plastics processing [2]. The polymers that are widely used in the food packaging industry are petroleum-based polymers, e.g., polyolefins (PO), polyethylene (PE), polypropylene (PP), polyethylene terephthalate (PET), polycarbonate (PC), polyethylene naphthalate (PEN), polystyrene (PS), polyvinyl chloride (PVC), ethylene vinyl alcohol (EVOH), and polyamide (PA) [3]. In particular, PE, PP, and PET enjoy great interest in the food packaging industry due to possessing good humidity and chemical resistance, low gas permeability, low density, high stability, and desired tensile strength and flexibility. Thus, these materials are role models for obtaining modern packaging in the food industry [3,4].

However, they are non-biodegradable and, in most cases, unsuitable for recycling. In effect, the production and use of non-biodegradable petroleum-based polymers contribute to worldwide ecological problems and health risks. Recently, 58.8 million tons of plastic was produced in Europe in 2022 [5]. This amount could increase to 1124 million tons of plastic material by 2050, considering that the use of plastic materials will be comparable to current levels [6]. Multilayer packaging with different types of combined plastic is the most dangerous for the environment and the most challenging to recycle. Although they provide good gas barrier and mechanical properties, they certainly cannot be recycled. For instance, multilayer packaging with polyolefins, PET, and PA cannot be recycled due to different recycling methods for polyolefins, PET, and PA. In this situation, recycling would be difficult to apply and unprofitable [7]. In addition, food packaging with plastic additives, such as phthalates, persists permanently even in recycled plastic. Phthalates, due to their low molecular weight and common migration to food or water, not only cause serious ecological issues but also lead to several health problems [8]. On the other hand, although recyclable materials seem promising for future food packaging strategies, the recycled polymer material itself may show different properties compared to the virgin material [9,10]. For instance, Müller et al. [11] reported a 30% drop in average molar mass and a 50% drop in break stress for recycled PET compared to the virgin material.

Since 2022, the European Union (EU) has implemented regulations regarding reducing single-use plastics in food packaging to decrease the environmental impact of disposable plastics. Recently, the food packaging industry has focused on producing more sustainable, healthy, and eco-friendly alternatives [12]. One of the choices is biodegradable polymers, both natural and synthetic, which degrade as a result of biological agent activity. Biopolymers are derived from biomass or microorganisms, mostly from agricultural and marine sources. Their leading representatives are polysaccharides and proteins, e.g., polyhydroxyalkanoate (PHA), polyhydroxybutyrate (PHB), and poly(lactide) (PLA) [4]. The complete biodegradation rate of PLA under controlled industrial composting conditions takes place between 90 and 120 days, while the biodegradation of PLA under home composting conditions does not take place [13]. The biodegradation rate of PHB is c.a. 90 days in organic media [14].

Biodegradable polymers could introduce additional functions for food packaging, e.g., hydrogel-based materials can easily absorb exudates from food products, prolonging the shelf life of the food [9]. Hydrogels are interesting materials that easily retain and absorb large amounts of water in their three-dimensional structure [15]. Hydrogel-based biopolymer packaging could also be easily functionalized by incorporating antimicrobial or antioxidant agents to minimize food spoilage or remove reactive oxygen species (ROS) [2]. Additionally, from an ecological point of view, the biodegradation rate of hydrogels is also impressive in comparison to traditional biodegradable polymers. For instance, sodium alginate (SA) hydrogel-based food packaging completely degraded after 12 days in a simulated body fluid (SBF) solution [16], while polyvinylpyrrolidone/carboxymethylcellulose hydrogel film completely degraded after 35 days under soil burial conditions [17]. Since such materials are biocompatible and biodegradable polymers, they are great candidates for replacing petroleum-based materials and a promising candidate for solving environmental problems.

On the other hand, these biopolymers usually possess poor barriers to water vapor, weak mechanical properties, and thermal stability in comparison to conventional plastic food packaging [18]. Moreover, their processing properties, chemical resistance, and the versatility of their physical properties are more limited than their fossil fuel counterparts. Another limitation is biopolymer degradation, which occurs relatively quickly in comparison to most expectations and needs of food packaging [19]. Finally, in multilayered packaging, the possibility of contamination of recyclable plastics with biodegradable residuals is significant [19,20].

Plenty of research must be done to design and fabricate bio-based food packaging that overcomes the above-mentioned issues. The food packaging and materials science industries are working on composites and biopolymer functionalization methods to obtain properties that match petroleum-based materials and simultaneously decrease their environmental impact.

Thus, this review thoroughly discusses current advances in designing hydrogel-based biodegradable materials for food packaging. In this respect, the innovative potential of hydrogel-based food packaging towards improved functioning, i.e., their use in active packaging, intelligent packaging, and a combination of these technologies, are reviewed. In this respect, the pros, cons, and future perspectives of materials dedicated to modern food packaging are thoroughly discussed. We emphasize discussion on the challenge of obtaining the required mechanical properties and discuss the options that allow this problem to be overcome.

## 2. Current Requirements for Food Packaging

For many years, the primary requirement imposed on food packaging was to contain and protect food from the external environment and mechanical, biological, and chemical damage, as well as to provide information on shelf life, ingredients, or nutrition [21]. However, such traditional food packaging has many significant drawbacks. Most of the materials used for traditional food packaging are non-biodegradable, leading to a worldwide impact on environmental pollution [22,23]. The current requirements for modern food packaging are much broader than those imposed on traditional packaging, as shown in Figure 1.

Besides traditional functions, food packaging should provide more properties to satisfy customers, industries, and legislation regarding food packaging production and the subsequent disposal of waste. The rising need for safer food has led to designing and applying a new type of packaging that extends its functionality. Active (AP) and intelligent packaging (IP) are two types of food packaging that could meet these expectations (Figure 2).

Aside from the functions of traditional packaging, AP is responsible for inhibiting microbial growth and activity, as well as food oxidation. This type of food packaging is functionalized with antimicrobial/antioxidant agents that inhibit microbial growth and thus extend the shelf life of food [9].

IP combines traditional food packaging systems with monitoring the changes in the system’s internal and external environmental conditions. In this field, time–temperature indicators of food history and sensors registering changes in gas, pH, humidity, or microbial activity are used. This approach improves the food’s safety and quality by sharing information on eventual issues [24]. IP aims to minimize food losses, provide information on food safety, and maintain the high quality of foods in whole food distribution until opening and consumption [2].

Besides improved functionality, modern food packaging should be made of materials that meet specific standards. Considering the current consciousness of consumers about sustainability problems, there is increasing interest in more eco-friendly food packaging in the global market [25]. In this regard, biomaterial-based packaging that characterizes biodegradability, recyclability, and the use of renewable resources is prioritized as sustainable packaging. Such products should provide consumers with maintained high food quality and uncomplicated handling and opening [23].

From an industry perspective, the scalability and decreased costs of producing and processing the material without sacrificing its properties are still demanded and finetuned. Further, the industry must take into account all of the legislations and restrictions imposed by the government to comply with circular economy principles and simultaneously decrease environmental impacts by providing reusability/biodegradation/recyclability. Lastly, the final product that fulfills all these requirements must be profitable for the industry [2].

Additionally, the life cycle assessment (LCA), i.e., an evaluation of the potentially negative influence of biomaterials on the environment, needs to be considered while designing novel food packaging. The LCA allows the assessment of product development in terms of socio-economic aspects, environmental impacts, and its usability and role in the field of conventional packaging materials. In this respect, the entire life cycle of the product, including resource production, design and production, usage, and waste management, needs to be taken into account [26]. Such studies could be essential in replacing currently used petroleum-based polymers as food packaging. For bio-based polymers, the LCA includes the separation of the polymer from other biomass components, an evaluation of the potential toxicity of used solvents, additives, and crosslinking agents, an evaluation of the amounts of used energy in the production process, and an assessment of polymer biodegradation [24]. Although biopolymers seem to be a decent alternative to petroleum-based polymers, the renewable agricultural feedstock used in biopolymer production, as well as the farming procedures used to grow the feedstock, could remain an environmental burden and consume high amounts of energy [27,28]. In the case of hydrogels, there is a lack of comprehensive LCA data enabling an understanding of their sustainability, which leaves a lot of room for improvement and further studies [2]

Beyond the requirements above, fundamental problems with the properties of food packaging materials still need to be solved. The perfect food packaging should characterize specific physical properties, such as providing an adequate gas/fluid barrier, high transparency, low weight, degradation ability/recyclability, and mechanical properties, i.e., strength, durability, and elasticity [25].

Since the gas/fluid barrier’s properties influence the shelf life of packaged products, they are crucial when designing food packaging. Most food products require stringent gas/fluid barrier properties. The exceptions are fruits and vegetables, where permeable conditions are needed to extend their shelf life. In this respect, the kinetics of water vapor permeation or moisture through the food packaging is determined via the water vapor transmission rate (WVTR). The gas barrier properties are calculated from O_2_, CO_2_, and N_2_ transmission through food as the gas transmission rate (GTR), using the partial pressure of the gas dependencies [29,30]. For typical commercially used food packaging, such as low-density polyethylene (LDPE), high-density polyethylene (HDPE), or PET, WVTR, and GTR, typical values are shown in Figure 3.

The perfect packaging material should characterize a lower rate of O_2_, CO_2_, and N_2_ permeability, decreased light transmission, and mechanical/thermal injury resistance. Otherwise, food spoilage occurs more quickly.

However, the biggest problem in designing new sustainable food packaging is finding a balance between adequate barrier and mechanical properties offered by fossil-based materials and biodegradability, which is a considerable advantage of biopolymers [33]. To overcome these issues, biopolymers could be mixed with other biopolymers or another biopolymer, or nano-additives such as nanoparticles (NPs) or nanofibers can be added [34,35]. Plenty of functionalization methods, e.g., adding fillers or crosslinkers, could be used by researchers, providing a wide range of possibilities in the formation of mechanically strengthened sustainable food packaging [36]. The mechanical properties required for food packaging materials and those currently obtained for new bio-based materials are discussed in Section 7.

## 3. Hydrogel-Based Films for Food Packaging

Hydrogels are tremendously promising materials in many fields of science and industry; this also applies to the food packaging industry. Hydrogels are polymers that can absorb and retain large amounts of water, forming hydrous 3D structures [37,38]. Due to this feature, hydrogels can absorb exudates from food and thus delay food spoilage [39,40]. In the food packaging industry, hydrogels are used as absorbent pads, biodegradable packaging, antimicrobial and antioxidant packaging, or colorimetric indicators [41].

There are several criteria for classifying hydrogels. They could be divided in terms of biodegradation ability, application, physical form, and responsiveness to stimuli according to the crosslinking method or their function [42].

The most fundamental and widespread hydrogel classification method is according to their origin. In this respect, synthetic, natural, or hybrid hydrogels could be distinguished. Examples of synthetic hydrogels are petroleum-derived polymers such as polyacrylamide (PAM), sodium polyacrylate (SPA), poly(acrylic acid) (PAA), polyethylene glycol (PEG), and polyvinylpyrrolidone (PVP) [43]. The natural hydrogels used for food packaging applications are proteins, e.g., gelatin (GEL), and polysaccharides, e.g., cellulose, starch, chitosan (CS), and SA [44,45]. They are derived from renewable sources, i.e., plants and animals [43,46]. The polymer, which consists of natural and synthetic hydrogels, e.g., PVP–carboxymethylcellulose (CMC), represents hybrid hydrogels [47,48]. The pros and cons of synthetic, natural, and hybrid hydrogels used in food packaging are discussed below.

### 3.1. Synthetic Hydrogel-Based Food Packaging

Synthetic compounds have been widely used in food packaging for many years. This is the effect of their numerous advantages, such as physical stability, mechanical strength, flexibility, and high biological and chemical resistance [9].

Other benefits are high repeatability and control over the hydrogel’s properties while designing and forming it, as well as better scalability, which, from the industry’s perspective, are crucial parameters for large-scale production [43].

The substantial drawback of synthetic hydrogels is the limited biodegradation ability of most of them [49]. There exist synthetic biodegradable polymers such as PAA, PLA, poly(glycolic acid) (PGA), 2-hydroxypropyl methacrylate (HPMA), or PAM. However, some of these, e.g., PAA, could migrate from packaging into food and have a carcinogenic effect on humans or release toxic residuals that could pose a tremendous danger to our health and planet [9,50,51]. This primary concern has pushed researchers and industries worldwide to look for more eco-friendly alternatives, such as bio-based hydrogel materials.

### 3.2. Natural Hydrogel-Based Food Packaging

Since natural hydrogel-based polymers are derived from renewable sources, they reduce carbon footprints and could solve environmental problems, causing an increased interest in this type of hydrogel [43]. Natural hydrogel-based materials offer many desired features, i.e., biodegradability, biocompatibility, sustainability, or adequate gas barriers for fruit/vegetable packaging [52]. Additionally, many natural hydrogels are considered to be safe and non-toxic when in contact with food. For instance, SA has been recognized by the Food and Drug Administration (FDA) as generally recognized as safe (GRAS), and the EFSA (European Food Safety Authority) approved its use and contact in specific doses with food [53,54]. They are noteworthy, decent alternatives to synthetic polymers.

Nevertheless, many challenges must be overcome. To industrialize such materials as food packaging, their poor mechanical properties, high brittleness/elasticity, and instability must be overcome [55]. For instance, Gong et al. [56] developed a self-assembled gallic acid (GA)/lysosome hydrogel with excellent microbial properties and biosafety that seemed promising for food packaging. However, the hydrogel was ineffective in forming films and possessed poor mechanical properties, which limited its practical use in this field. Another interesting material could be GEL, which is inexpensive, safe, and widely used in the food industry [57]. It has great film-forming and water-absorption abilities and is biodegradable. However, its poor mechanical properties, low thermal stability, and high water solubility make GEL on its own inappropriate for food packaging applications [58]. In other studies [59], starch-based hydrogel films showed good antimicrobial properties, but their high brittleness excluded them as potential food packaging. These issues could be overcome via adequate functionalization, e.g., by blending with other hydrogel-based polymers or other substances/crosslinking agents or adding nano-additives that could strengthen the hydrogel mechanically [15]. For instance, high water vapor transmission in alginates could be overcome by the addition of montmorillonite (MMT). In addition, the incorporation of this mineral also increases the efficacy of therapeutic and antimicrobial agents, as well as the mechanical properties, of alginate hydrogels [60].

Another issue of natural hydrogels is the vulnerability to microbial spoilage of some natural hydrogel-based films and compliance with FDA requirements. Natural hydrogels are especially sensitive to most sterilization methods, which can cause degradation of bonding in hydrogel structures, leading to a loss of material properties [61]. For instance, the studies conducted by Galante et al. [62] showed that conventional methods, such as sterilizing CS hydrogels using steam, heat, and gamma irradiation, were highly unsuitable. On the one hand, this issue could be partially overcome by functionalization with microbial agents. On the other hand, the FDA clearly stated that food products must be put into pre-sterilized containers with a pre-sterilized closure in sterile conditions and atmospheres [63]. Additionally, applying the two kinds of functionalization that improve mechanical and antimicrobial properties could be challenging in terms of the repeatability of results. The solution to this problem may be sterilization with the use of alternative methods. For instance, Labay et al. reported that low-pressure radio-frequency plasma sterilization did not change the physiochemical properties of the sodium alginate hydrogel [64].

The next challenge is scalability in industrial production, especially in a situation where the processing of bio-based hydrogels is not possible using several traditional methods. For instance, it is known that the highly efficient extrusion method allows for precise control of the production parameters. On the other hand, the shear stress accompanying this method could shorten macromolecules and thus influence the final properties of natural hydrogels [57]. On the other hand, many industry branches have increasingly used 3D printing over traditional methods due to the guarantee of food safety and quality [2]. According to up-to-date studies [65,66], 3D printing allows the extrusion of a higher fraction of active agents, i.e., antimicrobial substances or antioxidants enclosed into the hydrogel matrix, than cast cylinders, as well as with enhanced physical properties due to a more organized structure than the traditional solution casting method. Other advantages of this method are the printability of many bio-based hydrogels, precise structural control over production, enabling customization of printed structures and shapes, and low production costs [67].

### 3.3. Hybrid Hydrogel-Based Food Packaging

Multi-compound hybrid hydrogels are formed to overcome the main natural single-hydrogel limitations and introduce additional functionalities into the system. Hydrogel blending could provide antimicrobial properties, increase chemical resistance, and increase mechanical strength and flexibility [68]. Besides biodegradability and adequate physical properties, cutting-edge hydrogels for food packaging should offer more. Innovations in this field and the high demands of customers have led to the development of multifunctional packaging, enhancing preservation and maintaining food quality. An instance of such a hybrid hydrogel is reported by Wang et al. [69] in the form of a pH-sensitive CS−polyurethane (PU) hydrogel. Previously functionalized with amino groups and hydroxyl groups, CS provided good solubility in water, antimicrobial properties, biocompatibility, and biodegradability. In turn, PU increased the hydrogel’s mechanical strength and elasticity. In other studies [70], a bacterial cellulose (BC)/polyethyleneimine (PEI)-based hybrid hydrogel was developed as active food packaging. BC provided biocompatibility, biodegradation capabilities, and non-toxicity, while PEI, due to its polycationic nature, provided antibacterial properties. Together, they showed good thermal and mechanical properties and high processability.

More examples of hybrid hydrogel-based AP and IP, as well as innovations and challenges while designing such approaches, are described in Section 4.

## 4. Hydrogel-Based AP

Antimicrobial and antioxidant properties are two main features that characterize AP. The first is related to inhibiting or reducing the growth and spread of microorganisms and is obtained by incorporating antimicrobial agents, e.g., essential oils, probiotics, or micro- and nanoparticles. The second is related to the reduction of food oxidation by incorporating antioxidants, mostly natural extracts and essential oils [71]. In this respect, AP contains active components derived from biopolymers or attached antimicrobial/antioxidant compounds [72].

An attractive hydrogel-based film that can be used as AP was obtained by Mao et al. [73]. SA was incorporated with methal fenolic networks consisting of epigallocatechin gallate, iron (Fe), and thyme essential oil (SA/EGCG/TEON). In this composite, SA provided biodegradability, biocompatibility, and film-forming ability. EGCG/Fe/TEON increased the mechanical properties, elasticity, thermal stability, UV protection, and moisture and gas barriers of the composite. Additionally, the presence of TEON led to a 2-fold decrease in water vapor permeability (WVP) to 1.20 × 10^−11^ gm^−1^⋅s^−1^⋅Pa^−1^ and a 1.6-fold decrease in oxygen permeability (OP) to 2.25 × 10^−4^ gm^−2^⋅s^−1^. The SA/EGCG/TEON hydrogel film showed the sustained release of the essential oil, which provided prolonged antimicrobial properties against *E. coli* and *S. aureus*, as well as an antioxidant effect. The tensile strength and elongation at the break of the composite were 35 MPa and 15%, respectively. Although the hydrogel film has potential as AP, biodegradation tests and an estimation of the extended shelf life of the food product still need to be performed.

An interesting hydrogel approach that shows great promise in AP is stimuli-responsive hydrogels. Stimuli-responsive hydrogels are a class of polymers that enjoy great interest in many industries and also relate to food packaging [74]. They respond to stimuli from external environmental changes, i.e., temperature, pH, light, electricity, magnetic field, and so on [75]. Although stimuli-responsive hydrogels are mostly used in IP as real-time food quality sensors and food safety indicators, they could also play an important role in AP as antimicrobial/antioxidant-controlled delivery systems. For instance, an interesting study was conducted by Shaghaeh et al. [76] on stimuli-responsive synthetic hydrogel systems. A thermo/pH-responsive hydrogel made of (2,2,6,6-tetramethylpiperidin-1-yl)oxyl (TEMPO)-oxidized nanofibrillated cellulose and cationic-modified poly(N-isopropyl acrylamide-co-acrylamide) (TOCNF3/CPNIPAM-AM) was obtained and served as a natamycin (NA)-controlled delivery system in fruit packaging. pH changes and temperature increases in food environments promote the growth of microorganisms [77,78]. The composite showed swelling behavior due to the presence of PNIPAM below the lower critical solution temperature (LCST). Upon the temperature increasing above the LCST to c.a. 39 °C, due to the effect of hydrogen bond destruction, the hydrogel structure shrunk, resulting in NA release. The hydrogel’s responsiveness to a decrease in pH, corresponding to the beginning of food spoilage, was due to TOCNFs. NA was released from such a system for more than 5 days and its presence provided antimicrobial properties, showing a c.a. 15% inhibition of *S. aureus* and *E. coli* growth. The results also showed that the incorporation of TOCNFs increased the Young’s modulus of the hydrogel by almost 5-fold and increased the elongation at break by 2.1-fold. However, the Young’s modulus of this approach lay far below the values of commercial food packaging (Table 1). Therefore, an increase in Young’s modulus is expected before the commercialization of this composite as active food packaging. Additionally, in this system, approvable levels of acrylamide have to be imposed by the FDA.

Another example of a stimuli-responsive hydrogel-based film in AP is hybrid pH-sensitive eugenol functionalized chitosan/polyurethane hydrogels (CS/PU/EBs) [69]. Eugenol is an FDA-approved strong antimicrobial agent, which, when incorporated into a CS–polyurethane hydrogel, is more stable when in contact with the external environment, i.e., UV light, heating, and oxygen. Due to the presence of polyurethane, the hydrogel showed excellent tensile strength in the range of f 0.77–4.90 MPa and elongation at break in the range of 135–196%. These values are within the ranges of LDPE (Table 1), which is commonly used in food packaging.

The CS/PU/EBs composite also showed high thermal stability and high antimicrobial properties of up to 86% against *S. aureus* and *E. coli*. The CS/PU/EBs packaging showed a color change from yellow to green with the addition of Prussian blue, while the pH changed from 2 to 14.

Still, the balance between antimicrobial properties/extending the shelf life of the food and non-toxicity was challenging. The high amounts of eugenol groups, which are responsible for the powerful antimicrobial properties, resulted in toxic effects on L929 fibroblasts. Thus, their amounts in hydrogel films have to be tuned thoroughly, and the cytotoxicity of this approach should be precisely studied before industrialization.

Although pH-responsive hydrogels could detect the pH changes associated with early bacterial metabolism, the pH change could not be sufficient for detection until a substantial population has grown [79]. The detection effect could be enhanced by adding substances that are sensitive to the release of a volatile organic compound (VOC) to the hydrogel. According to Madivoli et al. [80], the incorporation of polydiacetylenes into the hydrogel matrices provided highly sensitive colorimetric probes to VOCs.

Among many biopolymers, cellulose-based hydrogel films enjoy great interest as AP due to their hydrophilicity, biodegradation ability, biocompatibility, and film-forming ability [81]. In this field, Aghajani et al. [82] designed hydroxylpropyl methylcellulose/Satureja khuzestanica essential oil (HPMC/SKEO) edible emulsion-based films as AP. HPMC served as an emulsifier and stabilizer, while SKEO provided antimicrobial, antifungal, and antioxidant properties. The hydrogel film showed strong antibacterial efficacy against *S. aureus*, i.e., the inhibition zone was 41.67 ± 1.53 mm on solid media, and the inhibitory effect lasted up to 30 days, as investigated by the disc diffusion method. Considering these properties, such an approach is a promising advancement for AP. On the other hand, the Young’s modulus, tensile strength, and elongation at break of these films were, respectively, in the ranges of c.a. 3–13 MPa, 19–44 MPa, and 10–17%, depending on the concentration of SKEO. The mechanical properties of HPMC/SKEO still need to be improved to fulfill the requirements of the food packaging industry.

Another cellulose-based hydrogel, i.e., CMC, is extensively studied and described in the current literature [81,83]. CMC is low-cost, derived from renewable sources, non-toxic, and shows better solubility and processability due to containing a large number of carboxyl groups in its main chain [83]. Although CMC-based hydrogels alone are characterized by low strains, poor mechanical strength, and quality loss under the effect of UV [84], proper CMC functionalization leads to outstanding properties that are highly desired from the perspective of food packaging applications. There are reports in the literature of CMC/PVA composites, which, due to the formation of hydrogen bonds between PVA and CMC, show better flexibility, transparency, and self-healing properties [71,83]. However, the mechanical properties of CMC/PVA hydrogel-based films are features that require improvement [85,86]. As an example, Gang et al. [86] reported blending CMC with PVA and pyrazinecarboxylic acid (POA) increased the tensile strength of the hydrogel by 1.3-fold and additionally provided antimicrobial properties, delaying browning and spoiling processes in bananas by up to 3 days. Although the composite film showed outstanding elasticity, i.e., elongation at break was in the range of c.a. 200–500%, the tensile strength of 5.5 kPa was not comparable in any way with commercially used food packaging. In other research, Yang et al. [87] obtained a CMC/PVA/tannic acid (TA)/ginkgo biloba leaf extract (EGBL) hydrogel film (CPTE). PVA stabilized the polymer chain, while TA served as an active antioxidant and antimicrobial agent. Additionally, due to the presence of hydroxyl and carboxyl groups, TA strengthens the hydrogel network mechanically. The presence of EGBL increased its film-forming ability, adhesion, gas barrier, and antimicrobial properties, as well as providing UV blockage. Additionally, in this hydrogel system, cutting-edge technology that solves hydrogel dehydration at ambient conditions has been implemented. Instead of using an aqueous solution, an organic solvent with a high boiling point was used. This not only solved the instability problem but also provided anti-freezing properties. The hydrogel system showed a UV-blocking effect, an antioxidation effect of c.a. 82%, an oxygen barrier of c.a. 8 cm^3^/(m^2^·24 h 0.1 MPa), and an extended shelf life of strawberries for the next 7 days. The total biodegradation of the hydrogel film occurred after c.a. 8 weeks.

Similarly, Zhao et al. [88] formed a CMC/PVA/TA hydrogel with the incorporation of polyethylenimine (PEI). In this composition, PEI played a supportive role in strengthening the hydrogel’s structure, while TA provided more crosslinking sites, thermal stability, UV resistance, and antimicrobial and antioxidant properties. Due to its powerful antimicrobial and antioxidant properties, the shelf lives of strawberries, cherries, and mangoes were extended for the next 7 days. The CMC/PVA/PEI/TA composite showed a tensile strength of c.a. 380 kPa and an elongation break of up to 400%.

Functionalization of CMC with minerals, e.g., MMT, homopolymers such as ε-poly-(l-lysine) (ε-PL), or other hydrogels, e.g., SA or natural extracts/anthocyanins such as purified thymus vulgaris leaf extract or aloe vera extract, results in better antimicrobial/antioxidant properties and could increase the elasticity of hydrogel-based films.

A considerable improvement in hydrogel properties, i.e., mechanical properties, thermal stability, or antimicrobial cues, could also be achieved via functionalization with nano-additives. Since there are many functional groups, such as hydroxyl, carboxyl, or aldehydes groups, in CMC’s structure, it could serve as an applicable matrix or encapsulation agent for the synthesis and conjugation of nanoparticles [89]. Indumathi et al. [90] loaded ZnO NPs into mahua oil-derived PU and CS films as active food packaging. In this composite, PU provided adequate flexibility, elasticity, and gas permeability, as well as thermal stability. CS was chosen due to its biocompatibility and antimicrobial and mechanical properties. According to the literature, e.g., [91,92], CS improved antimicrobial cues. This activity can be additionally improved by the addition of ZnO NPs as Zn^2+^ ions damage bacterial cell walls, which increases oxidative stress in the bacterial cell wall through the formation of active H_2_O_2_ molecules in the moisture environment. Not only did the addition of ZnO improve antimicrobial properties and increase the tensile strength of the PU/CS film by 1.6-fold, but also decreased the permeability of water and oxygen by 2.4-fold and 1.2-fold, respectively. Moreover, the whole composite showed excellent biodegradability, i.e., 86% after 28 days in the soil environment. The PU/CS/ZnO NPs system served as active food packaging for vegetables, which extended the shelf life of carrots for an extra 9 days. In comparison to polyester commercial foil, such an approach efficiently reduced the microbial activity of *S. aureus* and *E. coli*, although its elasticity was too low.

Similar studies used ZnO nanoparticles to obtain active packaging, such as the one reported by Kurabetta et al. [93]. The hydrogel AP film consisted of methylcellulose/gallic acid and ZnO NPs (MC/GA/ZnO NPs) and was obtained via the casting method. In this AP, gallic acid improved the water vapor barrier of MC and provided antioxidant properties. The ZnO NPs provided antimicrobial properties, high stability, a great gas/water barrier, and improved mechanical properties, as well as absorbing abundant amounts of UV light. The hydrogel film showed good compatibility between components as an effect of intermolecular interactions, as demonstrated by the FTIR and SEM methods. The WVP of the MC/GA/ZnO NPs hydrogel film was decreased by c.a. 2-fold in comparison to pure MC and was 9.5 × 10^−7^ (g/m h atm). The hydrogel film showed an excellent antioxidant effect of 94.5% and antimicrobial properties against *B. subtilis*, *S. aureus*, *E. coli*, *P. aeruginosa*, and *C. albicans*. Additionally, the AP film increased the shelf life of tomatoes by up to 27 days. After 20 days, the hydrogel film degraded by c.a. 60% in the soil burial test. The addition of ZnO NPs increased the tensile strength and Young’s modulus by 1.7-fold and 1.8-fold, respectively, making the strength of this approach comparable to commercially used food packaging.

In other studies conducted by Zafar et al. [94], a CMC/gGEL hydrogel loaded with ZnO NPs (CMC/GEL/ZnO NPs) was fabricated as active food packaging. The CMC/GEL/ZnO hydrogel showed thermal stability up to 100 °C, good water permeability of 3.1 g/m day atm, an antioxidant effect of 86%, and strong antimicrobial properties against many strains of bacteria, such as *S. aureus*, *B. subtilis*, *L. monocytogenes*, *E. aerogenase*, *E. coli*, and *B. bronchiseptica*. The composite showed a remarkable Young’s modulus of 0.765 GPa, a tensile strength of 45 MPa, and elongation at break of 10%, which could be comparable with the values of some of the petroleum-based polymers used in food packaging (Table 1).

Zhang et al. [84] formed a CMC/PVA hydrogel film enriched with copper sulfide nanoparticles (CuS NPs) as AP, with a long antimicrobial effect. CMC served as an encapsulation and stabilization agent in the synthesis of nanoparticles, while PVA conjugated CMC with CuS NPs. CuS NPs provide long-term antimicrobial properties. The CuS NPs inhibited the growth and spread of *S. aureus* and *E. coli* with 99% efficacy, showing excellent antimicrobial properties for up to 10 days. Those properties allowed the shelf life of strawberries to be extended for more than 6 days. The composite degraded by c.a. 30% after 20 days. The CMC/PVA/CuS NPs composite material showed great thermal stability up to 160 °C, adequate oxygen transmittance of 0.03 cc/m^2^/day, water permeability of 163.3 g/m^2^/day, and excellent elasticity, i.e., the elongation break was c.a. 180%.

Another hydrogel-based film was designed by Sagar et al. [95]. This approach consists of GEL/CMC, PVA, and SiO_2_NPs. The hydrogel system was crosslinked using tetraethoxysilane (TEOS) using air-dry casting. The addition of SiO_2_ NPs increased thermal stability, as well as the mechanical properties of the hydrogel system. This approach showed tensile strength and an elongation at break of 6.8 MPa and 118%, respectively, which lay in the ranges of commercial packaging (Table 1 and Table 3). Moreover, the system efficiently provides antioxidant and antimicrobial properties against *E. coli* and *B. cereus* as a result of the hydrophilic, oleophobic, and cationic nature of GEL/CMC/PVA/SiO_2_ NPs, which inhibits microorganism activity. Although further studies on gas/water permeability, whether this approach extends the shelf life of food products, environmental impacts, and market feasibility need to be performed, GEL/CMC/PVA/SiO_2_ NPs show great potential as AP.

Hydrogel-based film functionalization with nano-additives could also take place with the use of nanofibers. Pirsa et al. [96] fabricated a CMC/nanofiber cellulose/potassium permanganate (CMC/NFC/KMnO_4_) hydrogel film as AP for banana fruits. The KMnO_4_, due to its oxidizing properties, served as a humidity/ethylene absorbent, extending the shelf life of bananas. The CMC/NFC/KMnO_4_ film showed increased thermal stability and delayed the bananas’ ripening by up to 30 days by absorbing ethylene and humidity. In this composite, NFC provided the fibrous structure, increased the tensile modulus by 1.26-fold, and increased the tensile strength of the film by 2.3-fold. Nevertheless, due to the strong electrostatic NFC–CMC interactions, the NFC addition decreased the elongation break of the foil by 2-fold. Still, the obtained values are within the ranges of mechanical properties characteristic of commercial food packaging. Thus, this approach shows huge potential for active food packaging.

There are not many reports in the literature on the functionalization of CMC hydrogel-based food packaging films with nanofibers. This could be the effect of difficulties in balancing between providing excellent elasticity, usually by using non-degradable polymers, and biodegradability, which is a crucial ecological aspect. On the one hand, hydrogel functionalization with fibers using polymers such as PE, PP, and PA could provide outstanding elasticity (Table 1) but at the expense of biodegradability. On the other hand, hydrogel functionalization with biodegradable but brittle polymers such as PLA, PHB, and other biodegradable polymers [97] (Table 2) can, similarly to NFC, increase the tensile modulus and strength but reduce the elongation at break [98]. To overcome this problem, an adequate plasticizer, e.g., PEG, polyurethanes, soya bean oil, or other fillers such as organic metal frameworks, could be used for biopolymer functionalization using nanofibers. For instance, Khaturia et al. [99] increased the elongation at break of PLA by 5-fold through functionalization with Cu3BTC2 organic metal framework crystals. Thus, potential hydrogel-based film functionalization with bio-based nanofibers is a subject that has room to maneuver for scientists [97].

Although nanomaterials significantly improve the thermal stability, barrier, and mechanical and antimicrobial properties of hydrogel-based AP, the risk of their migration into food has to be considered. The toxicity of the NPs depends on their size, as smaller particles have a higher toxicity [100]. This results from enhanced surface reactivity, greater particle–biological system interactions, and the ability to cross biological barriers, which could lead to more toxic outcomes. Additionally, according to recent studies, the risk of migration of metal-based NPs increases in acidic environments and ethanol [101,102]. For instance, Rapa et al. [102] reported that ZnO NPs incorporated into PLA film migrated up to 3 mg/kg in acetic acid and up to 0.2 mg/kg in ethanol. However, this amount is within the allowable limit of NP migration according to the European Food Safety Authority (EFSA). Regulation (EU) No 10/2011 on plastic materials and articles intended to come into contact with food has established an allowable limit of migration for non-genotoxic substances in the range of 0.05 and 60 mg/kg for food or food simulants.

However, it should be taken into account that before industrialization, food packaging with nanomaterials needs comprehensive characterization, migration studies, and toxicological tests according to the EFSA’s guidelines. In the United States, the FDA requires providing data demonstrating the safety of nanomaterials used in contact with food via the Food Contact Notification (FCN) process. In this respect, the composition, potential migration into food, and toxicological tests of materials are required [100].

## 5. Hydrogel-Based IP

In addition to its traditional function as packaging, IP, or intelligent packaging, allows for the assessment of food safety, quality, and condition by monitoring particular parameters during production, transportation, selling, and purchasing [24,103,104]. The collected data provide important information on packaged food, reduce the amount of food poisoning and the common problem of wasting food, and improve food logistics [105].

IP systems could be classified in terms of their function and working principle into time–temperature and freshness indicators, as well as radio-frequency identification tags (Figure 4) [23].

### 5.1. Time–Temperature Indicators (TTIs)

The time–temperature indicator (TTI) is one of the approaches used by IP, which tracks the temperature history from food preparation and packaging through to transportation, storage, and opening [24]. The TTI evaluates whether the recommended storage temperature is accurately provided throughout the entire food supply chain. Since temperature maintenance is especially important for meat, fish, and dairy, the TTI is especially dedicated to these products [106]. Temperature increases up to certain values favor microbial multiplication and growth, leading to a shorter shelf life of the food.

The time–temperature effect is obtained via irreversible physical changes, mechanical deformation, or biological composition changes, resulting in a color change in the packaging [107,108]. The microorganisms’ growth results in decreasing pH levels in the fruit food environment to 4.5–5, CO_2_ release, and water evaporation. However, the release of total volatile basic nitrogen (TVB-N) corresponding to meat spoilage increases the pH in the food environment [109]. Thus, pH and temperature responsiveness are crucial features while designing IP.

TTI pH indicators bring many advantages and are a noteworthy decent approach in designing intelligent food packaging. Currently, many types of pH indicators are commonly used, i.e., chemical, physical, enzymatic, biological, and others.

The first type provides colorimetric changes as an effect of chemical reactions such as polymerization, photochromic reactions, or oxidation/reduction [110,111]. Polymerization occurs between monomers and acetylene groups, a photochromic reaction involves a photochromic substance that is activated via certain light wavelengths as an effect of a reverse reaction, while oxidation reactions show colorimetric changes as an effect of redox reactions [112,113].

The second type of TTI, i.e., physical, is based on changes in physical properties. In this respect, diffusion-based, nano- and microparticle-based, and electronic approaches are used [114]. The diffusion-based TTI uses the diffusion reaction of colored fatty acid esters along a porous material, which depends on temperature changes. The time–temperature dependence is measured as the distance between the advancing diffusion front from the beginning of the indicator [115]. The diffusion-based TTI also uses microorganisms or enzymes, which break upon freezing and thawing, releasing a colorful liquid as a result of increases with increasing temperature. For instance, isopropyl palmitate was used as a TTI indicator of threshold microbial growth in non-pasteurized angelica juice due to handling under non-standard temperatures [116]. Diffusion of isopropyl palmitate of more than 7.0 mm was a threshold point for microbial spoilage of the juice in temperatures above 13.5 °C. Although this type of TTI is suitable for a wide range of temperatures and its production is inexpensive and easy, a colored material exudation could influence the accuracy of the measurement [24].

An enzymatic TTI seems to be more accurate in that matter due to its sensibility to temperature changes and higher accuracy than diffusion approaches. The working principle of an enzymatic TTI uses the hydrolysis reaction of the enzyme with the substrate, which leads to a change in color. The rate of the reaction is conditioned by the time and temperature, so the color change shows the overall effect of the time and temperature. This allows the remaining shelf life of the food products to be dynamically displayed. An interesting enzymatic TTI was obtained by Meng et al. [117]. The direct enzymatic glucoamylase microcapsule/amylase/iodine/SA/CS solid TTI indicated the time and temperature dependence in spoiling pork meat. This TTI system is based on the enzymatic reaction between glucoamylase and amylase, which, after the addition of iodine, results in a change in color. The role of SA is to form microcapsules loaded with glucoamylase, which, in the presence of calcium chloride, results in fast gelation. CS provided the stability and sustained release of microcapsules. The TT history of the meat was evaluated by the color change of the TTI from mazarine to colorless, which corresponds to the fresh and spoiled states, respectively. The system detected the time of meat spoilage after a certain threshold of fatty acid deterioration was reached, which occurred after 130 h while stored at 4 °C, 65 h while stored at 15 °C, and 19 h while stored at 25 °C. However, the literature reports some limitations of enzymatic TTIs, such as instability and high costs [23].

Biological TTIs are based on the activity of fungi and bacteria. Such an approach consists of microorganisms, i.e., yeasts, bacteria strains such as *Lactobacillus* or *Streptococcus*, and colorimetric indicators, e.g., bromocresol green and methyl red. As a result of the anaerobic respiration of yeasts, the intensity of which depends on temperature, the pH of the environment changes to acidic through the production and accumulation of lactic acid, which causes the color change of the pH indicator. Biological TTIs, especially those based on lactic acid bacteria, are ecologically friendly by using inorganic media for strain growth and provide accurate analysis, a strong color change, and low-cost production. TRACEO^®^ and eO^®^ (both were developed and commercialized by Cryolog Clock-T°, Nantes, France) are commercial biological TTIs available on the market.

Nevertheless, in these approaches, environmental factors such as humidity, pH, temperature, and the occurrence of antimicrobials or other types of microbes could influence uncontrolled microbial growth, leading to customers being misled regarding the actual food’s quality and safety [117]. Additionally, most of the biological indicators are reproducible but only under controlled conditions. Testing these indicators in dynamic and real-life supply chains brings unpredictable responses regarding food spoilage as a result of unstable temperature and humidity conditions [118]. Other TTI approaches are those based on indirect detection. For instance, Pereira et al. [119] obtained indirect pH-sensitive CS/PVA/red cabbage-derived anthocyanin as a TTI, characterizing pH changes in the food while it is stored at higher temperatures for a certain period of time. The composite provided a color change in the pH range of 1–12 from red to green, respectively. Nevertheless, the Young’s modulus, tensile strength, and elongation at break were 3.53 MPa, 9.8 MPa, and 26.8%, respectively, which apparently is insufficient from the perspective of the food packaging industry. Such an approach could either be a hydrogel patch, i.e., an additive for currently existing food packaging, or that mechanical properties need to be improved in future studies.

### 5.2. Radio-Frequency Identification Systems (RFID)

Another type of IP approach incorporates RFID into the packaging. This enables food tracking during transportation and unloading and reduces the chances of theft. Such an approach could also monitor the freshness of the packaged food. Athauda et al. [103] obtained a CS/PEG-based hydrogel equipped with chip-less RFID resonators working at an ultra-wide band (UWB) of 3–7 GHz as a pH sensor for IP. In this composite, CS provides pH sensitivity, while PEG increases the mechanical properties, i.e., Young’s modulus and tensile strength, of the hydrogel system. CS’s pH sensitivity was registered as the change in its dielectric properties in the pH range of 4–10 as a result of hydrogel swelling and deswelling using microwave radiation. Thus, such an approach could be used as a pH sensor working in the UWB for smart food packaging applications, as long as only one acidic or alkaline condition is measured, and not both simultaneously.

Although hydrogel-based films are interesting approaches for smart packaging development, there are only few publications on hydrogel-based approaches incorporated with RFID [103,108]. Combining hydrogels with RFID could provide improved sensor performance and convenient usage. Thus, it could be an attractive future perspective for conducting more studies in this field.

### 5.3. Freshness Indicators

Recently, the development of freshness indicators based on colorimetric pH sensor films for applications in IP is a cutting-edge topic. They usually take the form of pads or tags that are incorporated into the food packaging. While designing such an approach, it is extremely important to use non-toxic and safe hydrogel compounds and colorimetric agents [120,121]. Such pH indicators change color upon pH changes, providing clear information to consumers on foods’ freshness and quality. In this field, synthetic dyes, such as bromothymol blue and methyl red, natural water-soluble anthocyanins, or flavonoids derived from plants are widely used [122,123]. Their mechanisms of action are based on pH changes caused by the dissolution of chemicals such as CO_2_ or volatile nitrogen compounds in water [124]. Providing an aqueous environment through the use of hydrogels plays an important role here, increasing the sensitivity of these types of freshness indicators [125]. An example of using synthetic dyes was reported by Lu et al. [123], who obtained a bagasse nanocellulose-based hydrogel for monitoring the freshness of chicken meat. The hydrogel was obtained by TEMPO oxidation of cellulose fibers, and subsequent chemical crosslinking with the Zn^2+^ ions. The hydrogel was a carrier of the pH-sensitive dye bromothymol blue/methyl red and served as an absorbent of CO_2_, which improved the sensitivity of the indicator. The hydrogel showed a visible optical change in color from green to red, while the pH changed from 9 to 3 after 4 days of being in contact with chicken breasts stored at 4 °C. However, the presence of pH-sensitive dyes reduced the strength of the crosslinked hydrogel. This is most likely the effect of the partial leaching of Zn^2+^ ions from the hydrogel system network during the dyes’ incorporation process.

To overcome this drawback, natural dyes such as anthocyanins could be conjugated to hydrogels [126]. According to some publications, anthocyanins could improve mechanical and barrier properties in relation to the water vapor of the hydrogel matrix [126].

The change in color of anthocyanins under pH changes occurs as an effect of their structural transformations [127]. Anthocyanins such as basil extract, potato anthocyanin, grape skin powder, fenugreek seed extract, alizarin, curcumin, betalain, shikonin, cinnamon essential oil, or flavonoids, e.g., the eucalyptus flavonoid, have attracted great interest in recent times [120,128]. The incorporation of natural colorimetric agents into the food packaging film provides real information on the shelf life of the food. In order to detect the food’s freshness, there are registered quality and quantity changes in the concentration of the substance that is responsible for food spoilage [88]. For instance, during the meat spoilage process, the meat’s proteins are colonized by bacteria, which release TVB-N compounds such as ammonia, dimethylammonium, and trimethylamine, resulting in pH changes in the surroundings [120]. This alteration in pH could be detected through the color change of a natural colorimetric agent incorporated into a hydrogel film [129]. For instance, Silva-Corrêa et al. [130] designed starch/PVA/vanillin (VN) as a colorimetric indicator of chicken meat freshness. Vanillin is a benzaldehyde that serves as a colorimetric sensor and chemosensor against Gram-positive and Gram-negative bacterial strains such as *B. subtilis*, *S. enteritidis*, and *E. coli*. The results of the food monitoring test showed a transparent hydrogel film containing a high concentration of vanillin, which changed color to brown upon the occurrence of bacteria strains such as *S. enteritidis* and *E. coli*, whose activity increased the pH of the meat after 10 days. Such an approach is an attractive colorimetric indicator for IP.

In other studies, Alizadeh-Sani et al. [126] developed a methylcellulose/chitosan nanofiber (MC/CSNF) hydrogel film loaded with barberry anthocyanin (BA) as a pH-responsive food freshness indicator. The BA was compatible with the composite film, increased the elastic modulus by 1.4, and improved the water vapor barrier of the composite. Most importantly, it provided a clear color change of the composite film from red to pale peach and yellow upon a pH change from 1 to 14 and the appearance of ammonia gas.

An interesting and prospective approach was also designed by Jiang et al. [131], who obtained PVA hydrogel patches loaded with curcumin. Curcumin previously functionalized upconversion fluorescent nanoprobes (UCNPs) obtained with the use of the fluorescence resonance energy transfer mechanism. This approach exhibited an extremely sensitive response to biogenic amines. The capturing of biogenic amines by UCNPs is based on proton transfer induced by H leads providing the change of diketone groups into enolate ions. The effect of this reaction is the absorption of non-radiative energy at 540 nm and the subsequent triggering of fluorescence quenching. This approach showed excellent detection of biogenic amines at the detection level of 2.73 μm, manifested by the color changing from green to red. Additionally, hydrogel pads incorporated with UCNPs were characterized by good stability for c.a. 2 h, repeatability, and the detection of food freshness without interference. Undoubtedly, an advantage of this approach is its reversibility. After providing non-alkali conditions and rinsing with water, the amine is rinsed out of hydrogel pads and UCNPs, making the system reusable, which is important from a practical and ecological point of view. In future studies, such systems will be joined with the color recognition system in smartphones to analyze the signals in the data and provide quantitative analysis. Since such an approach offers real-time monitoring of food freshness and provides an adequate food safety environment, it is a good direction for future research on food freshness indicators.

Another example of obtaining pH-sensitive hydrogel/anthocyanin systems for monitoring the freshness of food, e.g., shrimps, was described by Tang et al. [132]. In this study, the colorimetric indicator consisted of a PVA/SA hydrogel, aramid nanofibers (ANFs), and an anthocyanin extract from purple sweet potatoes (PSPE). In this approach, PVA/SA served as a biodegradable, water-soluble, and inexpensive hydrogel matrix. The presence of short ANFs of 5–10 μm in length increased the mechanical properties and provided structural stability for the hydrogel system as an effect of hydrogen bond network formation. ANFs were previously functionalized with H_2_SO_4_/HNO_3_ to increase the reactivity of the fibers as an effect of the occurrence of active groups of -NH_2_ and -COOH on the nanofibers’ surface. PSPE provided a visible colorimetric response from pink to green in the pH range of 2–12 and was sensitive to the presence of volatile ammonia. Additionally, the change in the indicator’s color was determined via a smartphone and special RGB software, which provided more accurate information on the shrimp’s freshness. Conjugation of the pH-sensitive indicator’s results of the TVB-N with an RGB-based response showed a satisfactory correlation of R^2^ > 0.9. These results show that such an approach could be accessible and useful to every person around the world and be applied to intelligent food packaging.

In other studies [133], a gellan gum (GG)/starch/red radish anthocyanin physically crosslinked hydrogel was formed as a pH indicator tag of milk and meat freshness. In this approach, GG provided the stronger incorporation of anthocyanins into the composite and regulated their release, while the addition of starch served as a porogen that modified the composite structure by phase separation. The porous structure of the hydrogel, with a pore size of c.a. 30 μm and a porosity of up to 75%, allowed gas transportation through the hydrogel matrix and increased the capture of food spoilage gases, increasing the sensitivity of this approach. In this approach, red radish anthocyanin changed the color from red to yellow upon a change in pH from 1 to 13, respectively. Although the incorporation of such intelligent, eco-friendly tags into food packaging could increase the functionality of conventional packaging, some limitations exist with such an approach. Large amounts of water carry the risk of contamination during the production process. Additionally, this feature and the natural origin of such an approach lead to faster biodegradation, which makes this product suitable for short-term use only, e.g., in meat, fish, or fruit/vegetable packaging. Moreover, the occurrence of starch significantly influences the color of the tags, increasing incorrect readings.

A prospective study was conducted by Wu et al. [134]. A CS/agarose/anthocyanin hydrogel obtained with the use of electrochemical deposition (writing) was obtained as a detector of fish freshness. The CS and agarose provide adequate mechanical properties as a result of hydrogen bonding, while anthocyanins serve as a colorimetric detector upon pH changes.

While the hydrogel system was placed on a Pt conductive surface that served as an anode, and stainless steel wire served as a cathode, a negative potential between the surface and the hydrogel occurred. By using stainless steel as a writing pen, a variety of information could be written with the anthocyanin on the hydrogel. By providing an adequate pH gradient in the cathode neighborhood, the electrodeposited anthocyanin in the CS/agarose hydrogel showed clear colorimetric information regarding pH changes.

The rise in pH near the wire led to the distinguished anthocyanin color change from red to blue. The hydrogel system after electrowiring showed a Young’s modulus of c.a. 3144 MPa, a tensile strength of 111 MPa, and an elongation at break of c.a. 30%. These properties are impressive in comparison with other hydrogel-based films and currently used petroleum-based polymers in the food packaging industry, and could be an interesting path for future research dedicated to intelligent food packaging.

Although anthocyanin incorporation has huge potential in use as freshness indicators, their storage conditions have to be considered while designing their applicability in the real world. Since anthocyanins are UV-sensitive, which leads to their degradation and thus loss of color, they need to be incorporated into a hydrogel film with good UV barrier properties [135]. Another crucial factor is the instability of anthocyanins at higher temperatures. According to Alighourchi et al. [136], the stability of anthocyanins is ensured at c.a. 5 °C, while storing anthocyanins at c.a. 40 °C could lead to rapid degradation. Anthocyanins may therefore be useful for food products that need to be stored in a refrigerator, such as meat, shrimp, fish, or some fruits and vegetables.

## 6. Combination of Active and Intelligent Hydrogel-Based Packaging

Currently, there is a strong trend of combining the properties of AP and IP. Although AP and IP alone bring many possibilities, providing both functionalities in one cutting-edge approach is tempting. This could be achieved by incorporating anthocyanins or nanoparticles with antimicrobial or antioxidant properties into current IP approaches, or by incorporating colorimetric indicators into AP [137,138].

An interesting study was conducted by Wang et al. [137], who obtained a pH-sensitive CS/black soybean seed coat extract (BSSCE) hydrogel film with antioxidant properties as IP. In this composite, CS was a hydrogel matrix with biodegradation capability, adequate mechanical properties, and gas permeability, while BSSCE provided pH sensitivity and antioxidant properties. The results showed that the presence of BSSCE in the hydrogel film increased tensile strength by 1.5-fold and elongation at break by 1.6-fold in the CS film, reaching a tensile strength in the range of 20–23 MPa and an elongation at break in the range of 65–74%. These values are within the range of commercial food packaging values.

The color change upon a certain pH occurred as an effect of the anthocyanin’s structural transformation. It was especially visible at pH 3–4, where BSSCE was bright red. At a pH of 5–7, it changed to violet, and turned blue at 8–10. The addition of BSSCE also increased the UV barrier of the film as a result of decreased UV transmittance, leading to a value of zero, and decreased the water vapor permeability of the hydrogel film by 1.3-fold. Additionally, the addition of BSSCE resulted in increased radical scavenging activities in the range of 34–52% due to the reaction of phenolic hydroxyl groups with free radicals. Such an approach could be suitable for active and intelligent food packaging.

The idea of intelligent/active hydrogel-based packaging seems to provide all of the needed functionalities of food packaging. However, tuning the chemical composition of hydrogel films that provide all of the properties required to implement them into the market is challenging. For instance, Qi et al. [139] obtained color-stable PVA/CS/purple tomato anthocyanin (PTA)/CaCl_2_ hydrogel-based films as AP/IP. The PVA/CS served as hydrogel-based matrices, while PTA provided colorimetric functionality and antioxidant properties. The CaCl_2_ served as a crosslinking agent and reinforced the physical properties. The antioxidant properties of the composite on DPPH free radicals reached c.a. 38% and antimicrobial properties against *E. coli* reached c.a. 25%. The pH response of the films occurred after 8 min from pink to gray, and ultimately to green, in the pH range of 2–3, 4–10, and 11–12, respectively, which was additionally confirmed by studies of monitoring pork meat freshness for 3 days. The tensile strength and elongation at break of PVA/CS/PTA/CaCl_2_ was c.a. 39 MPa and 56.5%, respectively. Although CaCl_2_ did not significantly increase the tensile strength, elongation at break was almost 2-fold higher after the addition of CaCl_2_s. Those mechanical properties meet the food packaging standards (Table 1 and Table 4). Still, the water vapor permeability showed that the presence of Ca^2+^ ions in PVA/CS/PTA/CaCl_2_ decreased the water barrier capacity of the hydrogel film, resulting in an increase in WVP as a result of the disruption of hydrogen bonding, limiting the number of interaction sites for water molecules.

Another challenge in designing intelligent/active food packaging, as with most of the natural bio-based hydrogel films, is the difficulty in obtaining adequate mechanical properties that are comparable with petroleum-based plastics. Such an example is the polyvinylpyrrolidone-carboxymethylcellulose-bacterial cellulose-guar gum (PVP-CMC-BC-GG)-based hydrogel films functionalized with clove/cinnamon essential oil (EO) to extend the shelf life of cheese [138]. In this composite, PVP-CMC-BC was a bio-based, biodegradable hydrogel matrix with good film-forming ability, while GG improved the mechanical properties, i.e., elasticity, tensile strength, and elongation at break, of the hydrogel system, making them suitable for the food packaging industry (Table 1 and Table 4) [140]. Additionally, anthocyanin pH-based stickers were immobilized to the hydrogel films to monitor the spoilage of the cheese. The EO functionalization provided antimicrobial properties against *Bacillus* and *Staphylococcus*, *E. coli* and *Klebsiellas*, and fungi of *Aspergillus* and *Candida*. The interaction between lactic acid and other organic acids in cheese with anthocyanins resulted in the change of the pH stickers’ color into pink, with more acidic conditions corresponding to cheese spoilage leading to more reddish stickers. The PVP-CMC-BC-GG hydrogel film showed 95% biodegradation after 60 days under moist soil conditions. The composite hydrogel film prolonged the shelf life of the cheese for up to 12 more days. Although most of the properties of the hydrogel were favorable, some of them negatively influenced the composite properties. The addition of clove/cinnamon EO enhanced Young’s modulus by 99% and tensile strength by 51% but decreased the elongation at break by c.a. 69%. Although the mechanical properties of PVP-CMC-BC-GG lay within the ranges of commercial food packaging (Table 1 and Table 4), functionalization with clove/cinnamon EO decreased the elasticity of the hydrogel film, limiting its potential use as food packaging.

The trials of providing additional components or other layers could increase the functionalities and physical properties of hydrogel-based packaging. However, such functionalization could also decrease other properties of the composite. For instance, Li et al. [127] obtained a PVA-CS/nano-ZnO/SA/cyanidin chloride pH-sensitive antimicrobial bilayer film. The first layer was composed of SA, while the second one consisted of PVA-CS obtained via the casting method and subsequently crosslinked with Ca^2+^ ions. In this composite, PVA-CS/SA are biodegradable, biocompatible polymers with good film-forming properties, while ZnO_2_ particles provide an antimicrobial agent. The presence of cyanidin chloride increased tensile strength and the UV barrier, as well as swelling ability, and decreased the WVP by 2-fold in comparison to the film without anthocyanin. The anthocyanin additionally provided a visible color change when the pH changed from 2 to 12 from red to yellow as a result of the structural changes of anthocyanins. The results showed a homogenous structure of bilayer films and enhanced physical properties, i.e., better light transmission and an improved UV barrier. Still, the antimicrobial effects against *E. coli* and *S. aureus* were comparable for the single and bilayer films. Additionally, better mechanical properties were obtained for single PVA-CS layered films than for PVA-CS/SA bilayer films. The tensile strength and elongation at break were, respectively, a c.a. 1.5-fold increase and a c.a. 5-fold increase for the PVA-CS single layer in comparison to the bilayer film. The authors reported that this was most likely the effect of decreased interactions between polymeric chains due to the presence of anthocyanins.

Finally, the most common problem in designing intelligent/active food packaging is providing simultaneous effective sensing and antimicrobial/antioxidant properties. Alpaslan et al. [141] designed a poly(gelatin-co-N,N-dimethyl acrylamide/citric acid/basilicum extract (DMAAm/CA/BE) hydrogel as IP for monitoring the pH changes of packaged food but also providing antimicrobial and antioxidant properties. In this composite, DMAAm decreased the high water solubility of GEL. Additionally, DMAAm, CA, and BE improved the mechanical properties of the GEL, obtaining a storage modulus of c.a. 500 MPa. BE served as an additional pigment, as well as an antimicrobial and antioxidant agent. The composite material shows visible differences in pH changes, with a bright color when the pH was near 1, and an intense dark color when the pH changed to 12. However, strong antimicrobial and antioxidant properties and BE co-pigmentation led to many difficulties in simultaneously sensing changes in pH and antimicrobial and antioxidant effects in packaged food.

In other studies, Mirzaei et al. [120] fabricated a hydrogel film composed of g κ-carrageenan loaded with either quercetin (CG/QUE) or eucalyptus leaf extract (CG/ELE) to prolong the shelf life and monitor the spoilage of chicken meat. The hydrogel film showed a high transparency of 90%, a high tensile strength of 13.2 MPa, a visible color change of QUE and ELE from bright yellow to dark yellow in the pH range of 1–12, and antimicrobial properties against *S. aureus*, *S. epidermidis*, *B. subtilis*, *E. coli*, *P. aeruginosa*, and *K. pneumonia*. However, these studies also revealed some differences in the pH measurement results of CG-ELE films depending on the eucalyptus species, harvesting season, region, or extraction method.

## 7. Comparison and Discussion of the Mechanical Properties of Conventional Non-Biodegradable and Degradable Food Packaging, and Currently Studied Approaches That Are Essential for Food Packaging

Since food packaging is exposed to various static and dynamic forces, depending on the type of food, food storage conditions, environments, and shelf life, as well as time and the form of transportation, different mechanical properties are required [142]. Young’s modulus, tensile strength, elongation at break, stiffness, and toughness are the most common parameters studied for materials dedicated to food packaging. Non-degradable petroleum-based plastics are still in common use as food packaging because of their excellent mechanical properties (Table 1). For instance, according to the literature [103,119], low-density polyethylene (LDPE) is one of the most attractive and hence popular food packaging materials (Table 1). In the case of biodegradable polymers, when designing cutting-edge food packaging, the main problem is providing a balance between adequate stiffness, toughness, and elasticity. Some of them are currently in common use as packing films, bags, or short-term packaging (Table 2). Although they are characterized by a wide range of Young’s modulus and tensile strength values that could be comparable to conventional plastics, they still lack adequate elasticity, i.e., biodegradable materials are characterized by significantly lower elongation at break values than conventional plastics (Table 1 and Table 2). For instance, PHB possesses good tensile strength but is very fragile. Another problem is the poor mechanical stability of biodegradable polymers, which could be overcome by the proper modifications [143]. In this respect, functionalization with micro- and nano-additives and composite formation, e.g., via polymerization, copolymerization, and blending, as well as by layer-by-layer film preparation, is commonly used to tune and improve the mechanical properties of biodegradable polymers.

Regarding most hydrogel-based films as AP, a significant disadvantage is their insufficient mechanical properties (Table 3). For instance, Shaghaleh et al. [76] obtained a TOCNF3/CPNIPAM-AM hydrogel film with an improved elasticity of 20% elongation at break, but Young’s modulus was c.a. 20,000 times lower than the lowest value in the Young’s modulus range for LDPE. In other studies [75], a CPTE hydrogel film was obtained with an excellent elongation at break of c.a. 210%, but Young’s modulus and the tensile strength were only 0.003 GPa and 0.09 MPa, respectively. Nonetheless, CMC-based hydrogel films possess mechanical properties that could be valuable from the perspective of food packaging. The Young’s modulus, tensile strength, and elongation at break were in the ranges of 0.1–0.8 GPa, 3.8–62 MPa, and 10–400%, respectively (Table 3). These values are in the range of LDPE, proving that these materials could be useful as active food packaging, especially from the perspective of mechanical properties.

**Table 1 polymers-17-01005-t001:** Mechanical properties of non-biodegradable food packaging and types of food packaging.

Material	Young’s Modulus (GPa)	Tensile Strength (MPa)	Elongation (%)	Type of Food Packaging	Refs.
Low-Density Polyethylene (LDPE)	0.11–0.45	2.7–200	100–956	food wrap bread bags	[103,144,145]
High-Density Polyethylene (HDPE)	0.6–1.1	17–45	10–1200	milk packaging beverage bottles	[144,145,146]
PP	1.1–1.5	31–43	500–650	yogurt containers, margarine tubs,	[144,145]
PET	2.8–4.1	48–270	45–100	beverage bottles, boil-in-bag pouches	[144,145]
PC	2.4	65.5	110	water bottles sterilizable baby bottles	[142,147]
PVA	-	37.5	126	food packaging film	[119,148]
Polyvinylidene Chloride (PVDC)	0.3–1.1	48–148	40–100	flexible monolayer film packaging	[142,144]
EVOH	2.1–2.6	59–77	230–380	multilayered films, e.g., milk packaging	[142,144]
PA	0.69–1.7	41–165	300–400	nylon bags	[144]
PS	-	30–60	-	hot beverage cups, takeaway boxes, egg cartons, meat trays	[142,145]
PCV	up to 4.1	10–55	14–450	cooking oil bottles, meat packaging	[144,145]
PTFE	-	7–28	-	-	[142]

**Table 2 polymers-17-01005-t002:** Mechanical properties of biodegradable polymers.

Material	Young’s Modulus (GPa)	Tensile Strength (MPa)	Elongation (%)	Type of Food Packaging	Refs.
PLA	2.3	32	5	packaging films, thermoformed cups short-shelf-life bottles	[142]
PCL	0.21–0.44	20–42	2.5–6	plastic bags	[142,149]
PHB	3.5–4	43	5–8	fruit and vegetable packaging	[142,149]
PHBV	0.2	25–40	13–20	packaging of solid food, e.g., trays or jars	[142,150,151]

**Table 3 polymers-17-01005-t003:** Properties of hydrogel-based AP.

Material	Young’s Modulus (GPa)	Tensile Strength (MPa)	Elongation (%)	Extending Shelf Life	Refs.
TOCNF3/CPNIPAM-AM	5.4 × 10^−6^	-	20.1	5 days	[76]
CS/PU/EBs	-	0.77–4.90	135–196	-	[69]
HPMC/SKEO	0.003–0.013	19–44	10–17	30 days	[82]
CPTE	0.003	0.09	c.a. 210	7 days	[83]
CMC/PVA/PEI/TA	-	0.38	400	7 days	[71]
CMC/MMT/ ε-PL	0.317	9.2	22.5	2 days	[152]
CMC/SA /*Thymus vulgaris* purified leaves extract (TVE).	-	62.2	61.1	25 days	[153]
CMC/PVA/Aloe vera	-	12.8	30	13 days	[154]
PU/CS/ZnO NPs	-	8.1	2	9 days	[90]
MC/GA/ZnO NPs	2.42	60.3	8	27 days	[93]
CMC/GEL/ZnO NPs	0.8	45	10	-	[94]
CMC/PVA/CuS NPs	0.03	0.034	180	6 days	[84]
GEL/CMC, PVA, and SiO_2_NPs	-	6.81	118	-	[95]
CMC/NFC/KMnO_4_	0.1	34	180	30 days	[96]

Considering hydrogel films as IP, the mechanical properties are not as significant as those of AP or traditional packaging. Many approaches for IP are not used as a whole package, but only a small part of it. For instance, for TTIs, hydrogel-based materials are usually in the form of pads or tags conjugated with conventional packaging and could differ mechanically from LDPE and HDPE, which are commercially used as food packaging [119].

In the case of active and intelligent food packaging, it appears to be different. Similar to AP, the mechanical properties of these approaches play a crucial role. Table 4 presents some of the examples, where Young’s modulus, tensile strength, and elongation at break were, respectively, in the ranges of 0.9–1.9 GPa, 13–39 MPa, and 5–74%. These values, especially Young’s modulus and tensile strength, correspond to those of LDPE or HDPE. Elasticity partially lies in the lower commercial packaging ranges, but it is the parameter that could be at least higher than 10%. This could be achieved by the addition of other substances, e.g., elastomers, to the hydrogel system, but other properties, such as biodegradability, adequate Young’s modulus and tensile strength, antimicrobial properties, and indicator sensitivity, need to be simultaneously kept.

**Table 4 polymers-17-01005-t004:** Properties of active and intelligent hydrogel-based packaging.

Material	Young’s Modulus (GPa)	Tensile Strength (MPa)	Elongation (%)	Extending Shelf Life	Refs.
PVP-CMC-BC-GG	0.94	25.9	21	15 days	[140]
PVP-CMC-BC-GG-EOs	1.4–1.87 GPa	c.a. 39 MPa	c.a. 6	12 days	[138]
CS/BSSCE		20–23 MPa	65–74	-	[137]
PVA/CS/PTA/CaCl2	-	39 MPa	56.5	-	[139]
PVA-CS/nano-ZnO/SA/cyanidin chloride	-	23–31 MPa	20–34	-	[139]
CG-ELE	-	13.2 MPa	5	-	[120]

## 8. Conclusions

Biopolymer-based packaging materials are at the forefront of active packaging research mostly due to their biodegradability, renewability, and biocompatibility. This is a result of emerging environmental problems coming from petroleum-based packaging and increasing consumer awareness. Another factor that has undoubtedly contributed to the field of food packaging is its improved functionality. Providing antimicrobial/antioxidant properties and detailed information on the food’s grade, characteristics, and safety plays a crucial role in modern food packaging approaches. Hydrogel-based films are great candidates, offering a number of advantages. First of all, they could absorb exudates from food. Additionally, bio-based hydrogel films are non-toxic, inexpensive, and show great biodegradation ability and processability. However, the achievement of adequate barriers, mechanical strength, and elasticity in hydrogel-based hydrogel films simultaneously is still challenging. Currently, the practical application of bio-based hydrogel films is limited, mainly covering the packaging of short-shelf-life food products such as fruits and vegetables, which require high humidity and respiration. In the case of long-shelf-life products such as dry pasta, conventional petroleum-based food packaging is still in common use due to providing an adequate barrier and mechanical properties simultaneously [155,156]. One of the directions toward overcoming the problem of insufficient mechanical properties is functionalization through polymer blending and the addition of nanoparticles/nanofibers. It was proven that hybrid hydrogels, such as hydrogels loaded with nanoparticles, show improved mechanical or thermal properties and could overcome the main limitations of hydrogel use in food packaging. Plenty of nanomaterials have already been tested as additives, improving the hydrogel’s water/gas barrier, thermal stability, and mechanical properties and increasing the hydrogel’s potential use in biomedical and food packaging. In this field, CuS, SiO_2_, and ZnO nanoparticles or cellulose nanofibers are commonly used.

Future studies on bio-based hydrogels dedicated to food packaging will most likely focus on their functionalization with nanofibers, which, according to the literature, has huge potential in overcoming issues related to the poor mechanical properties of bio-based hydrogels. Other future research directions might extend the functionalities of food packaging. In this respect, stimuli-responsive hydrogels, e.g., pH-responsive hydrogels, could be conjugated with RFID, in which hydrogel structural deformations could provide an accurate response regarding food quality and freshness. Finally, an important direction of research on innovative food packaging based on hydrogels is the creation of applications for mobile devices such as smartphones, enabling customers to obtain more accurate information about food products in a user-friendly way. For instance, conjugating a smartphone application with hydrogel colorimetric indicators into a QR code could identify the food product and provide information on its real freshness [157]. This approach could allow tracking and qualifying food products, while smartphone applications could evaluate food quality, freshness, and spoilage.

## Figures and Tables

**Figure 1 polymers-17-01005-f001:**
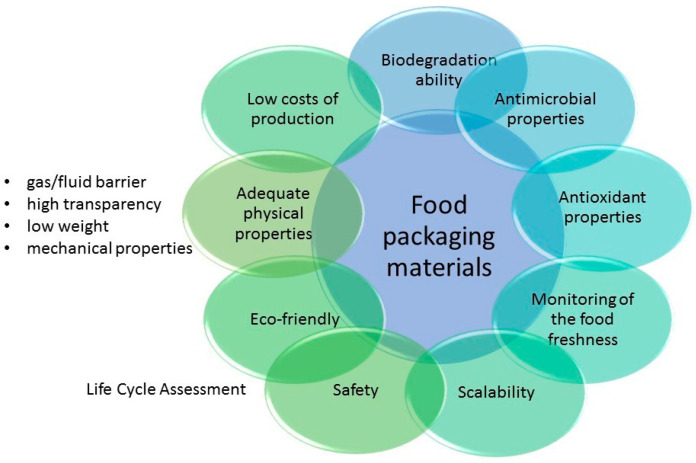
Requirements for modern food packaging materials.

**Figure 2 polymers-17-01005-f002:**
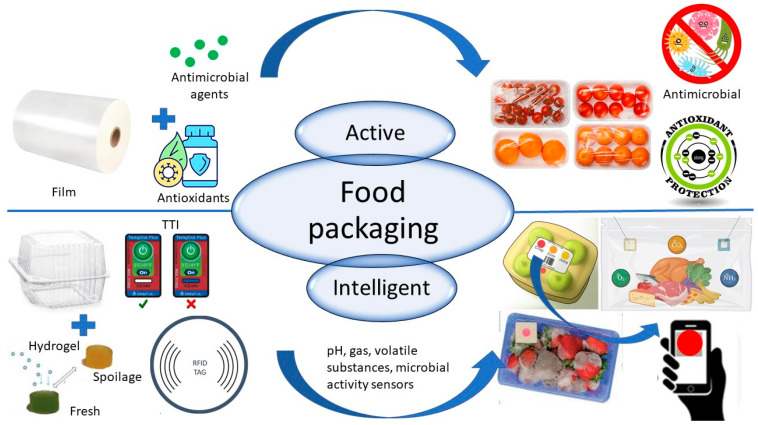
Modern technologies in food packaging.

**Figure 3 polymers-17-01005-f003:**
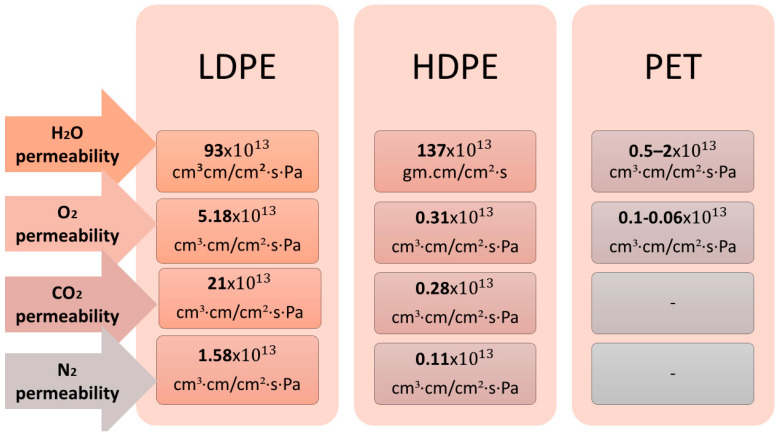
Permeability of water, O_2_, CO_2_, and N_2_ through LDPE, HDPE, and PET [31,32].

**Figure 4 polymers-17-01005-f004:**
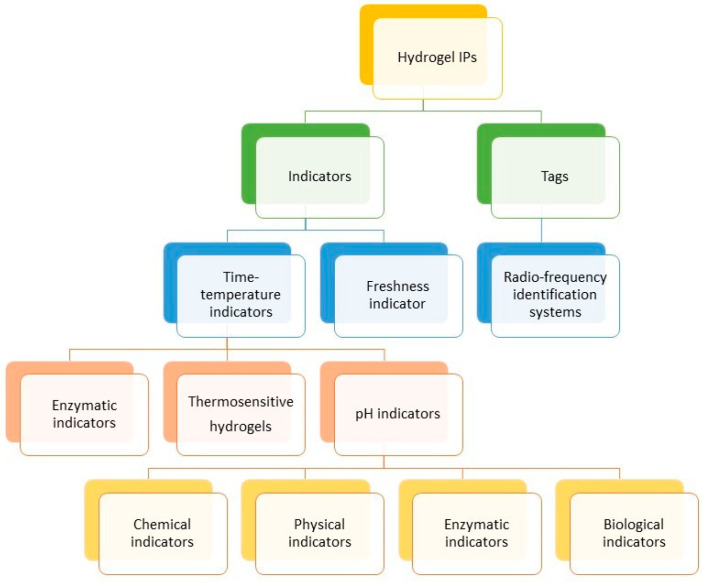
Classification of hydrogel IPs.

## Data Availability

No new data were created or analyzed in this study.

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
