# Peer review of "Hydrogel-Based Systems as Smart Food Packaging: A Review"

_polymers, 2025, doi:10.3390/polym17081005_

Round 1

Reviewer 1 Report

Comments and Suggestions for Authors

When reference is made to natural hydrogels, alginate is not mentioned. Alginate is one of the most important materials for the formation of hydrogels. Page 6 Lines 194-195.

The use of abbreviations throughout the text is not clear. Some are not specified and others are not in the right position. For example, CMC on page 9, line 344 or TA, line 361. These need to be reviewed...

In general, and as I have already mentioned, I miss more references throughout the text in relation to the use of alginate, as there are numerous articles published in relation to the use of this natural polymer in food packaging.

Author Response

Dear Referee,

Thank you for your valuable comments and suggestions. We will do our best to improve the quality of the manuscript. According to your comments, we have made corrections to the manuscript. Our answers and changes in the text are listed below.

1. When reference is made to natural hydrogels, alginate is not mentioned. Alginate is one of the most important materials for the formation of hydrogels. Page 6 Lines 194-195.

Response: The correction has been done.

2.  The use of abbreviations throughout the text is not clear. Some are not specified and others are not in the right position. For example, CMC on page 9, line 344 or TA, line 361. These need to be reviewed...

Response: We did our best to improve the clarity of abbreviations in the texts. The changes are highlighted in the text. For instance, the correction of TA abbreviation placement has been done. Still, in the authors' opinion, the CMC abbreviation is placed in the correct position in the text. For instance, the explanation of the CMC abbreviation of carboxymethylcellulose is placed on page 6, Line 216, which is before page 9, line 344.

3. In general, and as I have already mentioned, I miss more references throughout the text in relation to the use of alginate, as there are numerous articles published in relation to the use of this natural polymer in food packaging.

Response: We added in the text some examples of sodium alginate use as food packaging, for instance, Page 2, Lines 78-86, Page 7, Lines 238-242, Page 7, Lines 258-262, and Page 8, Lines 318-331.

Reviewer 2 Report

Comments and Suggestions for Authors

Hydrogel-based systems as smart food packaging. A review.

Please address the following major comments:

Lines 17–23: The abstract contains an informative overview of the scope and purpose of the review. But it could be more specific on some of the key shortcomings of hydrogel-based packaging that are currently preventing broad commercial adoption specifically its mechanical or barrier performance.

Lines 42–56: The paragraph discussing the recycling issues associated with multilayer plastics is effectively written. Comparative data on the biodegradability and degradation time periods of hydrogel-based films vs. traditional biodegradable polymers like PLA or PHB would further bolster the argument.

Lines 70–74: Since the primary purpose of the incorporation of hydrogels into packaging is in terms of the absorption of food exudates, have the authors evaluated in any previous work a quantitative correlation between the absorption capacity and the effect on microbial (e.g. yeast/mould) activity or, better, delayed spoilage of real food products? Such data would lend support to the functional claims made here.

Lines 103–107: Figure 1 describes the changing needs of the modern food packaging sector. Can the authors elaborate on how hydrogel-based approaches would be able to incorporate digital or smart features that are becoming increasingly requested for consumer-facing packaging (e.g., mobile app compatibility, QR-based tracking, etc.)?

140–149: Authors should be commended for including life cycle assessment (LCA) considerations. Can the authors mention any comparative LCA case studies for hydrogel biobased materials against standard bioplastics? This would make the position of hydrogels in the context of sustainability clearer.

Lines 237–247: Discussion around sterilization challenges of natural hydrogels, is very relevant. The authors have not yet encountered any promising non-thermal strategies of sterilization (for example, UV-C, ozone, or plasma) that may maintain hydrogel integrity while still conforming to food safety regulations? Here, just a mention would have added value.

Lines 289–312: Stimuli-responsive hydrogel systems are an exciting area of innovation. I would say the real world application of these data will be practically defined by metrics such as detection thresholds, lag times or sensitivity of these sensors to earlier microbial activities; this might be elaborated.

Lines 379–417: ZnO nanoparticles seem to play role on improving barrier and antifungal properties. If I may, considering the heightened scrutiny of nanomaterials in food-contact applications perhaps the authors could offer a brief treatment of the risks of migration of the nanoparticle into the food and relevant guidelines on safety (e.g. EFSA; FDA)?

Lines 486–542: The section on enzymatic and biological time-temperature indicators is very informative. The results would be more relevant if the applicability of these biological indicators would be tested for the real-life supply chain situation, where fluctuations in humidity and temperature may occur, and how reproducible and consistent these indicators are to detect spoilage?

607–619 In addition, the use of anthocyanins for monitoring freshness is indeed promising. However a response on their long term stability once exposed to light and in different temperature ranges(five °C to four °C) during storage and transport in real world packaging applications would aid in estimating their applicability in real world.

The use of a hydrogel film for both active and intelligent functions (lines 701–723) is quite interesting If so, the authors see any interesting reactions, antagonistic or synergistic between the antimicrobial agents and the pH-sensitive dye? Understanding this sort of interplay could inform better formulation strategies.

Author Response

Dear Referee,

Thank you for your valuable comments and suggestions. We will do our best to improve the quality of the manuscript. According to your comments, we have made corrections to the manuscript. Our answers and changes in the text are listed below.

1. Lines 17–23: The abstract contains an informative overview of the scope and purpose of the review. But it could be more specific on some of the key shortcomings of hydrogel-based packaging that are currently preventing broad commercial adoption specifically its mechanical or barrier performance.

Response: More specific information on hydrogel-based packaging limitations has been provided additionally in the abstract.

2. Lines 42–56: The paragraph discussing the recycling issues associated with multilayer plastics is effectively written. Comparative data on the biodegradability and degradation time periods of hydrogel-based films vs. traditional biodegradable polymers like PLA or PHB would further bolster the argument.

Response: The description of the biodegradability of traditional biodegradable polymers vs. hydrogels was placed in the introduction.

3. Lines 70–74: Since the primary purpose of the incorporation of hydrogels into packaging is in terms of the absorption of food exudates, have the authors evaluated in any previous work a quantitative correlation between the absorption capacity and the effect on microbial (e.g. yeast/mould) activity or, better, delayed spoilage of real food products? Such data would lend support to the functional claims made here.

Response: The authors haven’t found a quantitative correlation between absorption capacity and the effect of microbial activity. The absorption of food exudates by hydrogels is mentioned in many review papers and is explained as a reduction of the water activity of the product. Removing these fluids limits the available resources for microbial proliferation, which can inhibit the growth of microorganisms that cause spoilage and extend the shelf life of the food. Such information is available in: Singh AK, Itkor P, Lee YS. State-of-the-Art Insights and Potential Applications of Cellulose-Based Hydrogels in Food Packaging: Advances towards Sustainable Trends. Gels. 2023; 9(6):433. ; de Azeredo, H. M. (2013). Antimicrobial nanostructures in food packaging. Trends in food science & technology, 30(1), 56-69, or Nath, P.C.; Debnath, S.; Sridhar, K.; Inbaraj, B.S.; Nayak, P.K.; Sharma, M. A Comprehensive Review of Food Hydrogels: Principles, Formation Mechanisms, Microstructure, and Its Applications. Gels 2022, 9, 1.

4. Lines 103–107: Figure 1 describes the changing needs of the modern food packaging sector. Can the authors elaborate on how hydrogel-based approaches would be able to incorporate digital or smart features that are becoming increasingly requested for consumer-facing packaging (e.g., mobile app compatibility, QR-based tracking, etc.)?

Response: Stimuli-responsive hydrogels, e.g., pH-responsive, could be conjugated with RFID, in which hydrogel structural deformations are registered. Incorporation of information on hydrogel changes in its dielectric properties in the pH range of 4-10 as a result of hydrogel swelling and deswelling using microwave radiation could be registered and analyzed by incorporating with a smartphone application could provide an accurate response to food quality and freshness. Additionally, Hydrogel-based approaches could be incorporated into the digital or smart features by combining optical dyes for gas sensing with the hydrogel-based film and smartphone detection. Hydrogel-based approaches serve as biodegradable carriers of colorimetric dyes, which change the color of the pH indicator when exposed to ammonia. As mentioned in the conclusion, a smartphone application applied with a color map following hydrogel colorimetric changes could provide information on its real freshness. This approach could allow tracking and qualifying the freshness of food products. The example of such an approach is also written on Page 17, Lines 731-746.  

5. 140–149: Authors should be commended for including life cycle assessment (LCA) considerations. Can the authors mention any comparative LCA case studies for hydrogel biobased materials against standard bioplastics? This would make the position of hydrogels in the context of sustainability clearer.

Response: The short discussion on LCA studies on hydrogel has been added to the text (Lines 161-167).

6. Lines 237–247: Discussion around sterilization challenges of natural hydrogels is very relevant. The authors have not yet encountered any promising non-thermal strategies of sterilization (for example, UV-C, ozone, or plasma) that may maintain hydrogel integrity while still conforming to food safety regulations. Here, just a mention would have added value.

Response: The example of an alternative sterilization method suitable for hydrogels has been added to the text (Lines 274-277).

7. Lines 289–312: Stimuli-responsive hydrogel systems are an exciting area of innovation. I would say the real-world application of these data will be practically defined by metrics such as detection thresholds, lag times, or sensitivity of these sensors to earlier microbial activities; this might be elaborated.

Response:  Short discussion has been added to Page 9, Lines 376-381.

8. Lines 379–417: ZnO nanoparticles seem to play role on improving barrier and antifungal properties. If I may, considering the heightened scrutiny of nanomaterials in food-contact applications, perhaps the authors could offer a brief treatment of the risks of migration of the nanoparticle into the food and relevant guidelines on safety (e.g., EFSA; FDA)?

Response: We provided a discussion on nanomaterials migration into the food in Pages 12 and 13, Lines 529-548.

9. Lines 486–542: The section on enzymatic and biological time-temperature indicators is very informative. The results would be more relevant if the applicability of these biological indicators would be tested for the real-life supply chain situation, where fluctuations in humidity and temperature may occur, and how reproducible and consistent these indicators are to detect spoilage?

Response: The response was provided to the text in Page 15, Lines 631-634.

10. 607–619 In addition, the use of anthocyanins for monitoring freshness is indeed promising. However a response on their long term stability once exposed to light and in different temperature ranges(five °C to four °C) during storage and transport in real world packaging applications would aid in estimating their applicability in real world.

Response: The response was provided to the text in Page 18, Lines 778-787. 

11. The use of a hydrogel film for both active and intelligent functions (lines 701–723) is quite interesting If so, the authors see any interesting reactions, antagonistic or synergistic between the antimicrobial agents and the pH-sensitive dye? Understanding this sort of interplay could inform better formulation strategies.

Response: Since the active and intelligent hydrogel films as food packaging are a novelty, the interesting reactions between antimicrobial agents and pH-sensitive dyes have not been comprehensively studied yet.

Round 2

Reviewer 2 Report

Comments and Suggestions for Authors

The revisions have successfully addressed the issues raised in my review. I consider the manuscript suitable for publication in its current form.